# Functional Foods in Modern Nutrition Science: Mechanisms, Evidence, and Public Health Implications

**DOI:** 10.3390/nu17132153

**Published:** 2025-06-28

**Authors:** Mónika Fekete, Andrea Lehoczki, Agata Kryczyk-Poprawa, Virág Zábó, János Tamás Varga, Madarász Bálint, Vince Fazekas-Pongor, Tamás Csípő, Elżbieta Rząsa-Duran, Péter Varga

**Affiliations:** 1Institute of Preventive Medicine and Public Health, Faculty of Medicine, Semmelweis University, 1085 Budapest, Hungary; ceglediandi@freemail.hu (A.L.); zabo.virag@semmelweis.hu (V.Z.); balintmadarasz1993@gmail.com (M.B.); pongor.vince@semmelweis.hu (V.F.-P.); csipo.tamas@semmelweis.hu (T.C.); varga.peter@semmelweis.hu (P.V.); 2Health Sciences Division, Doctoral College, Semmelweis University, 1085 Budapest, Hungary; 3Fodor Center for Prevention and Healthy Aging, Semmelweis University, 1085 Budapest, Hungary; 4Department of Inorganic Chemistry and Pharmaceutical Analytics, Faculty of Pharmacy, Jagiellonian University Medical College, 31-008 Krakow, Poland; agata.kryczyk@uj.edu.pl; 5Department of Pulmonology, Semmelweis University, 1085 Budapest, Hungary; varga.janos.tamas@semmelweis.hu; 6Branch in Krakow-Hospital Pharmacy, Maria Sklodowska-Curie National Research Institute of Oncology, Garncarska 11 Str., 31-115 Krakow, Poland; duranela@poczta.onet.pl

**Keywords:** functional foods, nutrition and health, prevention of chronic diseases, probiotics, polyphenols, public health, nutrigenomics, nutritional science, prevention, health claims

## Abstract

In recent years, functional foods have garnered increasing scientific and public health interest due to their potential to confer physiological benefits beyond basic nutritional value. International bodies such as EFSA, FDA, and WHO define functional foods as those containing bioactive components that may contribute to the prevention and management of chronic non-communicable diseases, including cardiovascular disease, type 2 diabetes, and certain cancers. The evolving paradigm of “food as medicine” reflects a broader shift in nutritional science towards proactive, health-oriented dietary strategies. This article provides a comprehensive, interdisciplinary overview of functional foods by examining their biological mechanisms, clinical evidence, public health significance, regulatory frameworks, and future prospects—particularly in the context of advances in personalized nutrition and nutrigenomics. A thorough literature review was conducted, drawing from recent peer-reviewed studies and guidelines from key health authorities. The review highlights the roles of specific compounds such as probiotics and prebiotics in modulating the gut microbiome, flavonoids and polyphenols in anti-inflammatory processes, omega-3 fatty acids in cardiometabolic regulation, and vitamins and minerals in supporting immune function. While an expanding body of clinical trials and meta-analyses supports the health benefits of these compounds—including reductions in LDL cholesterol, improved insulin sensitivity, and mitigation of oxidative stress—the integration of functional foods into everyday diets remains challenging. Socioeconomic disparities and limited health literacy often impede their accessibility and widespread adoption in public health practice. Functional foods represent a promising component of prevention-focused modern healthcare. To maximize their impact, a coordinated, evidence-based approach is essential, involving collaboration among healthcare professionals, nutrition scientists, policymakers, and the food industry. Looking forward, innovations in artificial intelligence, microbiome research, and genomic technologies may unlock novel opportunities for the targeted and effective application of functional foods in population health.

## 1. Introduction

In the 21st century, one of the most pressing global public health challenges is the rising burden of non-communicable diseases (NCDs), including cardiovascular diseases [1,2,3], type 2 diabetes [4], obesity [5,6], and certain cancers [7,8]. Lifestyle-related factors—chiefly poor nutrition—play a pivotal role in the development of these chronic conditions [9,10,11,12,13,14]. Against this backdrop, the focus of nutrition science has shifted beyond basic sustenance toward disease prevention and health optimization [15,16,17,18,19,20]. In this context, the concept of functional foods has emerged as a bridge between traditional dietary habits and targeted health promotion [21].

NCDs are not only lifestyle-related—they are also fundamentally age-related diseases. Their pathogenesis is closely intertwined with the biological processes of aging [22], including cellular senescence [23,24,25,26,27,28,29,30,31,32], oxidative stress [33,34], epigenetic regulation [35,36,37,38,39,40], mitochondrial dysfunction [41,42,43,44,45,46,47], loss of proteostasis [48], chronic low-grade inflammation (“inflammaging”) [49,50,51], genomic instability [52,53], and altered neuroendocrine signaling [54]. Emerging evidence suggests that dietary factors exert significant modulatory effects on these aging mechanisms [55,56,57,58,59]. Nutrients and bioactive food components can influence the activity of key molecular pathways such as sirtuins, mTOR, AMPK, and Nrf2, modulate epigenetic patterns and mitochondrial health [60], and affect endocrine function and systemic inflammation [61]. Therefore, diet is not only a determinant of metabolic health but also a modifiable factor in shaping healthy aging trajectories [62,63,64,65,66,67].

Functional foods represent a promising strategy in addressing the complex challenge of non-communicable diseases, as they offer the potential to improve health outcomes and reduce disease risk through dietary diversification [68]. These are foods that, beyond their basic nutritional value, contain bioactive compounds capable of modulating physiological functions and contributing to the prevention of chronic diseases [69]. According to regulatory and scientific bodies such as the European Food Safety Authority (EFSA), the U.S. Food and Drug Administration (FDA), and the World Health Organization (WHO), functional foods may be either naturally occurring or intentionally modified to enhance their health-promoting properties [70]. Naturally functional foods include unprocessed fruits, nuts [71], vegetables, and whole grains [18] rich in essential nutrients, fiber [72], antioxidants, and phytochemicals [73]. In contrast, processed functional foods undergo deliberate compositional adjustments—such as enrichment, fortification, or reduction in certain ingredients—in order to increase their bioactive potential and support specific health outcomes [74]. Such modifications may involve the addition of beneficial components (e.g., omega-3 fatty acids, plant sterols, or probiotics) or a reduction in undesirable ones (e.g., saturated fats, sugars, or sodium). Functional food products can thus contain higher-than-average levels of health-enhancing ingredients or lower amounts of compounds associated with negative health impacts. In some cases, functional foods overlap with nutraceuticals—a category of products situated between conventional foods and pharmaceuticals—including concentrated extracts, dietary supplements, antioxidant-rich formulations, and encapsulated compounds such as fish oil or chitosan.

The growing popularity of the “food as medicine” paradigm reflects a broader shift in public and scientific discourse, where food is not merely a source of energy and essential nutrients but also a potential vehicle for disease prevention and therapeutic intervention [75]. This perspective is gaining importance as healthcare systems face increasing pressure from chronic disease management, while the public shows a growing preference for prevention-oriented and self-directed strategies for health maintenance [76]. However, the integration of functional foods into mainstream dietary habits remains limited by several factors. These include insufficient public understanding of scientific evidence, inconsistencies in the regulation of health claims, socioeconomic barriers to access, and a lack of standardized definitions and criteria for efficacy [77]. The gap between traditional dietary practices and evidence-based nutritional interventions continues to be a significant challenge for public health strategies.

Consequently, functional food research is not limited to nutritional science—it is a highly interdisciplinary domain that encompasses biochemistry, clinical medicine, food technology, regulatory policy, public health, and behavioral science [78]. Clinical trials and epidemiological studies increasingly support the health-promoting effects of specific functional components—such as probiotics [79], prebiotics [80], polyphenols, omega-3 fatty acids [81,82], vitamins [83], and minerals—on inflammation, metabolic health, gut microbiota composition [84], and immune modulation [85,86].

This paper aims to provide a comprehensive and critical overview of functional foods from an interdisciplinary perspective. It explores their mechanisms of action, clinical evidence, regulatory landscape, and implications for public health nutrition. In addition, the study highlights emerging directions such as nutrigenomics, personalized nutrition, and the integration of artificial intelligence into food science and health research. Ultimately, the goal is to contribute to scientifically ground public communication strategies that support the responsible inclusion of functional foods in everyday dietary practices.

## 2. Methods

In line with the objectives of this study, a narrative literature review was conducted, focusing on the scientific foundations, clinical evidence, public health relevance, regulatory context, and future perspectives of functional foods and their health effects. The review was carried out between November 2024 and May 2025 and followed the methodological steps outlined below.

### 2.1. Database and Source Selection

Relevant scientific literature was identified using several major international databases, including PubMed, Scopus, Web of Science, and ScienceDirect, along with official publications and guidelines from the EFSA, FDA, and WHO. The search focused on peer-reviewed articles published in the last ten years (2014–2025), with particular attention to randomized controlled trials (RCTs), meta-analyses, systematic reviews, and public health recommendations.

### 2.2. Keywords and Search Terms

The main search terms included the following: functional foods, bioactive compounds, chronic disease prevention, probiotics, polyphenols, nutrigenomics, personalized nutrition, public health, health claims, and regulatory frameworks. Boolean operators (AND, OR) were used to construct complex search queries.

#### Inclusion and Exclusion Criteria

Studies were included in the review if they

were based on human clinical research (animal studies were excluded);adhered to recognized methodological quality standards;investigated the physiological effects of specific functional foods or components;were published in English or Hungarian.

Studies were excluded if they

were speculative and lacked primary data;relied solely on in vitro or animal model data.

Data extraction and analysis

The following data were extracted from the included studies:the type of functional component investigated (e.g., probiotic, flavonoid);the duration and dosage of the intervention;characteristics of the study population;outcome measures evaluated (e.g., cholesterol levels, insulin sensitivity, microbiome composition, cognitive function, inflammatory markers, glycemic control, antioxidant status, immune response);statistical significance and effect sizes.

The extracted data were synthesized narratively, as the heterogeneity of the included studies (in terms of intervention types, populations, and methodologies) precluded the possibility of a quantitative meta-analysis. The results were interpreted within the broader contexts of public health, nutrition science, and regulatory policy.

The methodological foundation of this study is based on the structured presentation of the two main categories of functional foods. The analysis distinguishes between, on the one hand, natural functional foods, which inherently contain bioactive compounds beneficial to health (e.g., blueberries, broccoli, kefir), and, on the other hand, intentionally modified or functionalized foods, which acquire their functional properties through industrial processing and targeted enrichment (e.g., margarine fortified with omega-3 fatty acids, yogurts supplemented with pre- and probiotics).

A differentiated examination of these categories enables a clearer understanding that not all seemingly healthy foods qualify as functional, and not all functional foods are of natural origin. This distinction is based on classifications presented in the reviewed literature, ensuring that the categorization reflects the original research context and enhances transparency in the evaluation of functional foods. Such differentiation is essential for the objective assessment and scientific interpretation of functional food efficacy.

In accordance with the official definitions provided by the EFSA and the WHO, functional foods are defined as conventional food products that contain bioactive components capable of exerting beneficial effects on one or more target functions in the body beyond basic nutritional value, and are consumed as part of a regular diet—not in the form of pills or supplements [87,88,89]. This regulatory perspective helps delineate evidence-based functional foods from general health-promoting or fortified products.

Numerous natural functional foods—such as garlic, turmeric, green tea, and various berries—are well-documented for their positive health effects. Similarly, flaxseed, olive oil, nuts, fermented foods (e.g., natto, kimchi), pomegranate, and citrus fruits have been recognized for their roles in reducing inflammation, providing antioxidant protection, supporting cardiometabolic health, and maintaining gut microbiota balance. However, due to space constraints, this manuscript focuses on the most frequently studied intervention foods, rather than providing an exhaustive overview.

## 3. Physiological Effects and Preventive Potential

Functional foods contain biologically active components that, beyond providing essential nutrients, may exert targeted physiological effects that contribute to the maintenance of health and the prevention of chronic non-communicable diseases [87,88,89]. According to recent scientific evidence, the consumption of these foods can positively influence the human body through a variety of well-defined biological mechanisms [21,87,88,89,90].

One of the most frequently studied aspects of functional foods is their antioxidant capacity, which helps to neutralize reactive oxygen and nitrogen species, thereby reducing oxidative stress-induced cellular damage [91,92]. This mechanism is particularly important in the prevention of neurodegenerative diseases, cardiovascular conditions, and age-related physiological decline [93]. Compounds with antimutagenic properties—such as certain polyphenols, carotenoids, and folic acid—can mitigate the damaging effects of mutagenic agents on DNA, thus decreasing the risk of malignant transformation [94]. The anticarcinogenic potential of various bioactive substances has been confirmed in numerous in vitro and in vivo studies, particularly in the case of sulforaphane, isoflavones, and conjugated linoleic acid [95].

Functional foods containing probiotics and prebiotics can favorably modulate the composition of the gut microbiota, thereby supporting immune function, maintaining intestinal barrier integrity, and reducing the risk of inflammatory bowel diseases [96]. The prebiotic effect primarily enhances the growth of beneficial bacterial strains (e.g., *Bifidobacterium*, *Lactobacillus*), while probiotics contribute to immune modulation and pathogen exclusion through their direct microbiological activity [97].

Omega-3 fatty acids—especially eicosapentaenoic acid (EPA) and docosahexaenoic acid (DHA)—play a critical role in cardiovascular protection [98,99,100,101,102]. Their effects include lowering triglyceride levels, inhibiting the synthesis of inflammatory mediators, and improving endothelial function [98]. Similarly, soluble dietary fibers such as β-glucans, which are mainly found in oats and barley, help reduce LDL cholesterol levels, thereby lowering the risk of atherosclerosis [103,104,105,106].

Many functional foods also exhibit immunomodulatory effects, particularly those containing vitamins C, D [1,107,108,109], and E, as well as trace elements like zinc and selenium [100,110,111]. These compounds enhance immune cell activity, modulate inflammatory responses, and support effective host defense against infections. In addition to strengthening immune function, certain bioactive components possess neuroprotective and neuroregulatory properties [112]. Omega-3 fatty acids, B vitamins, and polyphenols can positively influence neurotransmission and reduce neuroinflammation, helping to preserve cognitive function and decrease the risk of neuropsychiatric disorders [113,114,115].

Phytochemicals—such as curcumin and sulforaphane—are also capable of exerting epigenetic effects by modulating gene expression through mechanisms such as DNA methylation, histone acetylation, and regulation of non-coding RNAs [116]. This opens new avenues in the personalized prevention of chronic diseases [117].

A distinct category within functional foods includes those aimed at the prevention or management of obesity [118]. These typically feature reduced energy content, such as low-sugar or low-fat formulations, which play a vital role in maintaining energy balance and preventing metabolic syndrome [119]. Moreover, sodium-reduced products are of particular importance in the population-level prevention of hypertension, especially in countries where daily salt intake exceeds the recommended levels [120].

In conclusion, functional foods exhibit a wide range of biological activities that support health maintenance and contribute to the prevention of lifestyle-related diseases. However, their use should be guided by scientifically validated evidence and applied within a well-defined nutritional context. This includes careful consideration of their mechanisms of action, effective dosages, bioavailability of active compounds, and the specific needs of the target population.

## 4. Clinical and Epidemiological Evidence on the Efficacy of Functional Foods

Over the past two decades, there has been a substantial increase in the number of randomized controlled trials and meta-analyses investigating the physiological effects and disease-preventive potential of functional foods. The growing body of clinical and epidemiological data convincingly supports the notion that foods containing specific bioactive components may play a significant role in the primary prevention of cardiometabolic diseases, type 2 diabetes mellitus (T2DM), metabolic syndrome, and certain types of cancer.

Functional foods are generally classified into three main categories:

### 4.1. Naturally Functional Foods

These are whole, minimally processed foods that inherently contain health-promoting, beneficial compounds. Examples include:Blueberries, which may positively influence cardiovascular and cognitive functions;Broccoli, which exhibits significant anti-carcinogenic and anti-inflammatory properties;Kefir, which supports gut microbiota balance and immune function through its probiotic content.

### 4.2. Intentionally Modified Functional Foods

This category includes foods whose composition has been deliberately altered through industrial processing to enhance their content of specific bioactive components such as nutrients, antioxidants, prebiotics, or probiotics. Representative examples include:Margarine enriched with omega-3 fatty acids (cardiovascular protection)Yogurt containing added prebiotics (support for gut microbiota)Micronutrient-fortified products (ensuring essential nutrient intake)Formulations enriched with polyphenols (enhanced antioxidant activity)

### 4.3. Functionalized Foods

A third emerging category includes conventional foods that are fortified during processing with functional ingredients—often plant-derived compounds like polyphenols—to enhance their antioxidant capacity or gut health benefits. While many findings in this area stem from in vitro studies, the potential for broader health applications is increasingly recognized. Examples include:Polyphenol-enriched baked products, showing increased antioxidant activityPasta fortified with carrot and olive leaf flour, resulting in higher polyphenol content and enhanced antioxidant capacityYogurt with added mango peel powder, demonstrating improved in vitro prebiotic effects

There is a growing body of scientific evidence supporting the preventive and therapeutic application of such foods. However, the objective evaluation of their efficacy requires rigorous and transparent research practices. In the following sections, we provide a detailed summary of relevant studies, including tabulated data for clearer comparison.

## 5. Clinical and Epidemiological Evidence

The health-promoting effects of functional foods are increasingly supported by scientific research. Randomized controlled trials and systematic reviews provide reliable insights into how specific food components affect human physiology [121,122,123,124]. These studies reveal the ways in which such foods influence inflammatory processes, metabolism, hormonal balance, and cardiovascular function. Long-term prospective cohort studies offer particularly strong evidence for the preventive role of functional foods in chronic, non-communicable diseases such as metabolic syndrome, type 2 diabetes, and certain types of cancer [125,126]. This growing body of data serves as a critical foundation for developing dietary guidelines and shaping public health strategies.

Natural functional foods are traditional foods that not only provide essential nutrients but are also proven to have beneficial effects on various physiological processes in the body, thereby contributing to the prevention and treatment of chronic, non-communicable diseases [127]. These foods naturally contain bioactive compounds such as antioxidants, polyphenols, flavonoids, phytochemicals, prebiotics, and probiotics, which together support metabolism, immune responses, and the regulation of inflammatory processes [128].

Natural functional foods include, among others, fresh fruits and vegetables, whole grains, legumes, and certain fermented dairy products [129]. Through their complex nutrient and phytochemical profiles, they can influence health at multiple levels. For example, certain fruits, such as berries, are particularly rich in polyphenols, known for their antioxidant and anti-inflammatory properties [130]. Vegetables, such as cruciferous vegetables, contain compounds [glucosinolates, isothiocyanates] that have cell-protective and potentially cancer-preventive effects [131].

Whole grains—such as oats, barley, buckwheat, or quinoa—are rich in dietary fiber, B vitamins, magnesium, and phytic acid, which together contribute to stabilizing blood sugar levels, lowering cholesterol, and maintaining the balance of the gut microbiome [132]. Additionally, fermented dairy products like kefir, due to their probiotic content, can promote gut health and optimize the immune response [133].

Numerous clinical and epidemiological studies confirm that the regular consumption of these foods can have a positive impact on cardiovascular health, metabolic functions, and insulin sensitivity, as well as cognitive and mental well-being [134,135]. Therefore, natural functional foods are not only part of daily nutrient intake but can actively contribute to health maintenance and the reduction of disease risk. The following tables illustrate the health effects of natural functional foods, with clinical research clearly supporting their positive impacts on various populations and health conditions (Table 1 and Table 2). However, further long-term RCT studies will help deepen the understanding of the mechanisms and enable the targeted application of functional foods in the prevention and treatment of chronic diseases.

## 6. Multifaceted Health Benefits of Kefir: Cardiovascular, Metabolic, Gut Microbiota, and Cognitive Perspectives

Recent studies have consistently shown that kefir, a probiotic-fermented dairy product, offers a wide range of health benefits, particularly in the areas of cardiovascular risk reduction, metabolic health improvement, modulation of gut microbiota, and enhancement of cognitive function. These findings highlight kefir as a promising functional food with therapeutic potential.

Cardiovascular risk and inflammation: Numerous studies have demonstrated that kefir consumption can positively influence cardiovascular risk factors. In a study by Bellikci-Koyu et al. [136], participants consuming kefir experienced significant reductions in LDL-C levels, systolic and diastolic blood pressure, and pro-inflammatory cytokines such as TNF-α and IL-6. Bellikci-Koyu et al. [137] reported similar improvements in fasting insulin levels and blood pressure, although no significant between-group differences were observed compared to control groups. Additionally, Yılmaz et al. [138] observed significant reductions in erythrocyte sedimentation rate and *C*-reactive protein (CRP) in patients with Crohn’s disease after 4 weeks of kefir consumption, along with improvements in subjective well-being and a reduction in bloating.

Metabolic syndrome and insulin resistance: Kefir also appear to have beneficial effects on metabolic health. Alihosseini et al. [139] demonstrated that kefir supplementation significantly reduced the Homeostatic Model Assessment for Insulin Resistance (HOMA-IR), indicating improved insulin sensitivity in patients with type 2 diabetes. Additionally, both kefir and conventional fermented milk groups showed reductions in homocysteine levels. Bellikci-Koyu et al. [137] also reported significant decreases in fasting insulin, HOMA-IR, and inflammatory markers, though no statistically significant differences were found between the kefir and control groups. In a study by Fathi et al. [140], kefir intake led to reductions in total cholesterol (TC), LDL-C, and non-HDL-C, indicating a potential benefit in improving lipid profiles in overweight or obese individuals.

Gut microbiota modulation: The influence of kefir on gut microbiota composition has been widely studied, with positive results. Öneş et al. [141] reported increased microbial diversity, specifically an increase in *Akkermansia muciniphila* and *Faecalibacterium prausnitzii*, which are associated with improved metabolic and inflammatory status. Wang et al. [142] found that kefir consumption significantly increased *bifidobacteria* abundance in male subjects and improved the gut microbiota composition in female participants, with increases in total anaerobes and total bacterial counts. Additionally, Pražnikar et al. [143] observed significant reductions in serum zonulin levels following kefir consumption, indicating improved gut barrier function.

Cognitive function: Kefir consumption has also been associated with improvements in cognitive function, particularly in neurodegenerative diseases. In Ton et al. [144], patients with Alzheimer’s disease showed significant improvements in memory, visual–spatial abilities, and executive/language functions after 90 days of kefir supplementation. This was accompanied by reductions in inflammatory cytokines and oxidative stress markers, suggesting that kefir’s neuroprotective effects may be mediated through its anti-inflammatory and antioxidant properties. Similarly, Noori et al. [145] reported improvements in depression and appetite regulation following the consumption of probiotic-fortified kefir (*Lactobacillus helveticus* and *Bifidobacterium longum*), though no significant changes in oxidative stress or inflammatory markers were observed. The evidence presented suggests that kefir, as a probiotic dairy product, offers a range of health benefits, including a reduction in cardiovascular risk, improvement of metabolic health, modulation of gut microbiota, and enhancement of cognitive function. These effects are likely attributable to kefir’s anti-inflammatory, antioxidant, and microbiome-modulating properties. Given its potential to improve multiple aspects of health, kefir may represent a valuable dietary intervention for the prevention and management of various chronic conditions. Further research, particularly long-term, large-scale studies, is necessary to fully understand the mechanisms behind these effects and to optimize the application of kefir in clinical practice. The table below (Table 1) summarizes the results of randomized controlled trials investigating the health effects of kefir and other fermented dairy products in various populations.

## 7. Effects of Fruits, Vegetables, and Plant-Based Functional Foods on Cognitive and Cardiometabolic Parameters

Numerous clinical studies have demonstrated that regular consumption of fruits [176], vegetables, and their bioactive components—primarily polyphenols, dietary fibers, and vitamins—exerts beneficial effects on both cognitive functions and cardiometabolic parameters. Nilsson et al. [150], in a 16-week intervention study among elderly individuals, confirmed that increased intake of fruits and vegetables led to a significant anti-inflammatory effect, as evidenced by reductions in inflammatory markers, while physical activity levels remained unchanged. Similarly, studies by Carrillo [151] and Arcusa [152] reported that polyphenol-rich fruit and vegetable extracts significantly improved executive functions, short-term memory, and attention. These cognitive benefits were accompanied by reductions in oxidative stress markers, including oxidized LDL and CRP levels, alongside increased dopamine concentrations, supporting the biochemical basis for enhanced neural function.

Regarding the role of the microbiome, Kopf and colleagues [153] observed in overweight subjects that a six-week intake of whole grain cereals and vegetables resulted in marked decreases in inflammatory cytokines IL-6 and TNF-α, as well as increased gut microbial diversity, a key factor in anti-inflammatory mechanisms. Consistent with these findings, postprandial studies by Yuan [154] demonstrated that consumption of fiber-rich functional bread favorably modulated blood glucose levels, contributing to metabolic balance without adverse effects.

The cognitive and emotional benefits of plant-based diets were further substantiated by Campbell et al. [155] who showed that an eight-week whole-food plant-based dietary intervention in women with metastatic breast cancer improved cognitive functions, reduced fatigue, and enhanced emotional well-being, with excellent adherence.

Berries and berry-derived juices, including grape, cherry, blueberry, and citrus fruits, have also been shown in multiple studies to confer cognitive advantages, particularly in memory, reaction time, and attention, while reducing blood pressure and inflammatory markers. Lamport and colleagues [156] reported improved cerebral blood flow and cognitive test performance following grape and citrus juice consumption, whereas Haskell-Ramsay’s study [157] confirmed rapid cognitive improvements after a single dose of grape juice. Whyte [172] documented preserved cognitive performance under fatigue conditions following a berry smoothie intake. Nilsson [158] highlighted that the combined presence of polyphenols and fibers favorably modified lipid profiles by reducing total and LDL cholesterol levels while improving working memory.

Cruciferous vegetables such as broccoli also exert positive effects on cardiovascular health: Connolly’s RCT [159] demonstrated reductions in systolic blood pressure, while Riso et al. [160] observed decreased CRP levels. Moreover, Navarro’s [161] investigations revealed that the anti-inflammatory effects were modulated by Glutathione S-transferase (GST) GSTM1/GSTT1 genotypes, underscoring the role of genetic factors in dietary responsiveness.

The significance of antioxidants and vitamins was emphasized in studies by Khan [162], where blackcurrant juice consumption significantly lowered oxidative stress markers and increased plasma vitamin C levels. Similarly, Duthie [163] reported elevated vitamin and carotenoid concentrations following 12 weeks of fruit and vegetable intake, although antioxidant capacity remained unchanged. Nadeem’s research [164] indicated a reduction in serum amyloid A [SAA] protein in the HDL fraction, potentially improving HDL’s anti-atherogenic function.

Functional fruit juices also show promise for cognitive maintenance. Siddarth [165] demonstrated that a 12-month pomegranate juice intervention preserved visuospatial learning in older adults, while Ahles’ study [166] with citrus- and pomegranate-based beverages improved handgrip strength and reduced oxidative stress markers. Kennedy’s research [167] on functional breakfast consumption revealed a lower insulin response, though no significant effects were noted on mood or memory.

Several clinical studies have confirmed the beneficial effects of plant-based foods rich in polyphenols and other bioactive compounds on cognitive performance, inflammatory markers, and metabolic parameters [58]. Kent et al. [168] investigated the effects of anthocyanin-rich cherry juice (200 mL/day for 12 weeks) in older adults (≥70 years) with mild to moderate dementia. The intervention led to significant improvements in verbal fluency (*p* = 0.014) and short- and long-term memory performance (*p* ≤ 0.014), along with a reduction in systolic blood pressure (*p* = 0.038). However, no significant changes were observed in CRP or IL-6 levels.

In an acute single-blind crossover study, Lamport et al. [169] examined the effect of a single 500 mL dose of flavanone-rich citrus juice (70.5 mg flavanones) in healthy young adults (*n* = 24). Two hours post-consumption, increased cerebral blood flow was observed in the right frontal gyrus, along with improved performance in the Digit Symbol Substitution Test (DSST), although no other cognitive improvements were reported.

Rosli et al. [170] conducted a 10-week intervention using polyphenol-rich tropical fruit juice in 31 middle-aged women with impaired cognitive function. Significant improvements were found in immediate verbal memory (RAVLT) and cognitive flexibility (CTMT Trail 4, *p* < 0.05). Additionally, urinary levels of thyroxine and 3-methyladenine—metabolites associated with cognitive performance—were elevated.

Saeidi et al. [171] assessed the combined effects of daily broccoli supplementation (10 g/day) and combined aerobic–resistance exercise training (CARET) over 12 weeks in men with type 2 diabetes. The intervention groups showed significant reductions in plasma dectin-1 levels (*p* < 0.05), along with improvements in insulin resistance and various cardiometabolic markers (*p* < 0.05).

Whyte et al. [172] evaluated the acute effects of a single 400 mL dose of a flavonoid-rich mixed-berry smoothie in healthy young adults. While the placebo group experienced a decline in cognitive performance due to fatigue, the berry group maintained accuracy and showed improved reaction times between 2 and 6 h post-consumption. Gouvarchinghaleh et al. [173] investigated the effects of a plant-based functional soup mixture in hospitalized COVID-19 patients. Daily consumption resulted in significant reductions in inflammatory cytokines (IL-1β, IL-6, IL-17, IL-10, TNF-α), D-dimer, blood urea nitrogen (BUN), and creatinine levels (*p* < 0.05). Notably, CRP and potassium levels improved more prominently in the control group.

Buckwheat-based products have also demonstrated beneficial health effects. Studies by Stokić [174] and Dinu [175] reported that regular consumption of buckwheat-containing foods significantly reduces blood lipid levels and oxidative stress markers, while concurrently enhancing antioxidant capacity.

Furthermore, regular intake of tomatoes and tomato-derived products, such as tomato juice, has been shown in clinical trials to exert a broad spectrum of physiological benefits. These include improved antioxidant status, attenuation of cardiometabolic risk factors, and favorable modulation of glucose metabolism. Based on these outcomes, tomato juice may be regarded as a functional food, a classification substantiated by a substantial body of clinical evidence [177,178,179,180,181,182,183,184,185,186].

In summary, the synergistic action of bioactive compounds found in fruits, vegetables, and other plant-based functional foods plays a pivotal role not only in preserving and enhancing cognitive performance, but also in reducing inflammatory and oxidative stress markers, lowering cardiometabolic risk, and optimizing key metabolic parameters. These findings reinforce the critical role of plant-based dietary patterns in health promotion and chronic disease prevention. The following table (Table 2) summarizes the findings of clinical studies on the health effects of fruits, vegetables, and other plant-based dietary interventions.

## 8. Health Benefits and Molecular Mechanisms of Blueberries

Scientific interest in the health benefits of blueberries (*Vaccinium corymbosum*, in particular) has markedly increased in recent years, largely due to the potential preventive and therapeutic roles of polyphenols found in the fruit—especially anthocyanins (ACNs) and other bioactive (poly)phenols [187,188,189]. These compounds contribute to cardiovascular health through various biological mechanisms [190]. A targeted search in medical databases using terms such as “blueberries and cognitive function,” “blueberries and inflammation/oxidative stress,” “blueberries and metabolic health,” and “blueberries and cardiovascular health” identified over 100 published clinical studies. Due to space limitations, this article does not provide a comprehensive review of these studies. Instead, we focus on the key molecular mechanisms through which blueberry (poly)phenols exert their effects on the pathophysiology of cardiovascular diseases. We specifically address their influence on inflammatory responses, oxidative stress, endothelial function, and glucose metabolism, as these are critical factors in reducing cardiometabolic risk.

### 8.1. Modulation of Inflammatory Processes

One of the most prominent mechanisms of action of blueberry polyphenols is the modulation of inflammatory responses [191]. A central player in inflammation is the nuclear factor kappa-light-chain-enhancer of activated B cells (NF-κB), a transcription factor activated by oxidative stress that promotes the expression of pro-inflammatory mediators, such as cytokines (TNF-α, IL-1α, IL-6), chemokines, and adhesion molecules [192]. Numerous in vitro and in vivo studies have demonstrated that anthocyanins found in blueberries can inhibit NF-κB activation and its translocation into the nucleus, thereby reducing the production of these inflammatory mediators [193,194,195]. Moreover, blueberry-derived compounds may increase the levels of anti-inflammatory molecules such as adiponectin and interleukin-10 (IL-10), further enhancing the anti-inflammatory response [196].

### 8.2. Reduction in Oxidative Stress

Oxidative stress plays a key role in the development of cardiovascular diseases [3,197,198,199,200]. Blueberry polyphenols exhibit potent antioxidant properties, partly through activation of the Nrf2/Keap1 signaling pathway [201]. Activation of nuclear factor erythroid 2–related factor 2 [202] promotes the expression of various antioxidant enzymes, including superoxide dismutase (SOD), glutathione peroxidase (GPX), and catalase, which are essential for neutralizing reactive oxygen species (ROS) [203]. In addition to upregulating endogenous antioxidant defense systems, blueberry bioactives are also capable of directly scavenging free radicals by donating electrons, thereby reducing their reactivity and harmful effects on cellular structures [204].

### 8.3. Improvement of Vascular Function

Blueberry polyphenols exert beneficial effects on vascular homeostasis, particularly through enhancement of endothelium-dependent vasodilation [205]. A key mechanism is the activation of endothelial nitric oxide synthase (eNOS), which increases the production of nitric oxide (NO) [206]. NO plays a crucial role in vascular tone regulation, endothelial health, and blood pressure control [207]. Furthermore, blueberry polyphenols may downregulate vasoconstrictive mediators, such as cyclooxygenase-2 (COX-2), and reduce oxidative stress, thereby supporting vascular function [193]. Recent findings also suggest that these compounds may stimulate glucagon-like peptide-1 (GLP-1) activity, which further enhances eNOS expression and NO synthesis, contributing to improved endothelial relaxation [193].

### 8.4. Regulation of Glucose Metabolism

Blueberry polyphenols also play a role in maintaining glucose homeostasis by improving insulin sensitivity and lowering blood glucose levels [208]. Both animal and human studies have shown that anthocyanins can improve glucose tolerance and promote cellular glucose uptake through upregulation of the glucose transporter GLUT-4 [209]. Additionally, these polyphenols may suppress hepatic gluconeogenesis by downregulating the expression of key enzymes such as glucose-6-phosphatase (G6Pase) and phosphoenolpyruvate carboxykinase (PEPCK) [210]. Blueberry compounds have also been shown to activate AMP-activated protein kinase (AMPK), a critical regulator of cellular energy balance and insulin sensitivity [211].

### 8.5. The Role of Blueberry Polyphenols in Neurodegenerative Disease Prevention

While the cardiometabolic benefits of blueberry polyphenols have been extensively studied, their potential neuroprotective effects in the context of neurodegenerative diseases have received increasing scientific attention. Neurodegenerative conditions are strongly associated with chronic inflammation, oxidative stress, and impaired neuronal signaling. Blueberry-derived polyphenols can cross the blood–brain barrier and exert direct effects on neural tissue, reducing oxidative damage and inflammatory signaling in the brain [212,213].

Clinical trials have shown that blueberry supplementation improves memory performance, working memory, and verbal learning in older adults [187,214,215]. These cognitive enhancements are thought to be mediated by increased brain-derived neurotrophic factor levels, synaptic plasticity, and neurogenesis [216]. In a randomized, double-blind, placebo-controlled trial, healthy older adults aged 60–75 years consumed one cup of freeze-dried blueberries [approximately 24 g] daily for 90 days, resulting in significant improvements in cognitive tests related to memory and executive function [187]. These improvements were associated with elevated levels of specific polyphenol metabolites in the bloodstream. In animal studies, blueberry extracts have been found to reduce β-amyloid deposition, a hallmark of Alzheimer’s pathology, and to improve cognitive performance and motor coordination [217,218].

Collectively, current findings support the hypothesis that polyphenol-rich foods such as blueberries may contribute to the prevention and adjunctive treatment of both neurodegenerative and cardiovascular diseases. The polyphenols present in blueberries exhibit complex biological activities, including the modulation of inflammation, oxidative stress, endothelial dysfunction, glucose metabolism, and neuronal signaling. These effects are particularly relevant in conditions such as Alzheimer’s disease, where blueberry-derived compounds can cross the blood–brain barrier and exert protective actions on neural tissue.

## 9. Intentionally Modified Functional Foods: Categories and Physiological Benefits

A key subgroup of functional foods comprises those products that are intentionally modified to enhance their nutritional value or to promote specific health-related outcomes. In the case of these so-called intentionally modified functional foods, the addition of bioactive components (e.g., phytosterols, omega-3 fatty acids, probiotics, micronutrients) serves not only to prevent deficiencies but is also aimed at specific preventive or therapeutic purposes—such as addressing cardiovascular diseases, dyslipidemia, or gut microbiota imbalances [219].

Dairy products, particularly yogurt, have received considerable attention in the food industry during the development of functional products. This is due to their widespread consumption, high biological value, ease of flavoring, and suitability as carriers for bioactive compounds. According to recent research, yogurts enriched with probiotics and micronutrients can serve not only to enhance nutrient content but also fulfill preventive and therapeutic roles [219]. Live microorganisms incorporated into functional yogurts—especially strains of *Lactobacillus* and *Bifidobacterium*—can restore and maintain gut microbiota balance, support the immune system, and positively influence digestion. Thanks to yogurt’s popularity, low cost, and adaptability to taste preferences, it represents an excellent base for the development of functional foods. In their study, Abdi-Moghadam et al. [219] highlighted that yogurts enriched with probiotics and micronutrients have been proven to reduce the incidence of diseases associated with nutritional deficiencies and exert beneficial physiological effects on the human body.

Prebiotic components, such as inulin, also play an important role in maintaining gut health. In a randomized, double-blind, crossover study, Russo et al. [220] demonstrated that pasta enriched with inulin significantly reduced intestinal permeability [measured by the lactulose-mannitol ratio] and zonulin levels, while increasing GLP-2 hormone levels. These findings suggest that prebiotics, including inulin, can effectively support the integrity of the intestinal epithelium, promote metabolic balance, and reduce the risk of inflammatory bowel diseases.

Fortified foods containing phytosterols (PS)—including plant sterols and stanols—are increasingly recognized for their LDL-C lowering effects in the context of cardiovascular disease prevention. A targeted meta-analysis by Fontané et al. [221], based on 125 randomized controlled trials, evaluated the LDL-C-lowering effects of these foods as well as influencing factors. The analysis showed an average LDL-C reduction of 0.55 mmol/L with phytosterol-enriched products, with a dose-dependent response. The most effective reduction occurred at daily intakes above 2.5 g PS. These results align with previous meta-analyses [222,223], which found LDL-C reductions between 0.34 and 0.41 mmol/L with similar doses.

The type of fortified food also plays a significant role: PS in “butter, margarine, and spreads” proved significantly more effective than in “bread, biscuits, and cereals,” likely due to lower doses and reduced bioavailability in solid, starchy matrices. In contrast, fat-based carriers such as margarine may improve PS absorption and thus enhance efficacy. The frequency (once vs. multiple times daily), timing with or without meals, and treatment duration did not significantly influence LDL-C changes. It is important to note, however, that LDL-C reduction appears within a few weeks and remains stable for at least three months. Clinically noteworthy is the finding that phytosterols further lowered LDL-C even when taken alongside statin therapy, making them especially beneficial for individuals who do not meet lipid targets with statins alone. As such, the European Society of Cardiology (ESC/EAS, 2019) guidelines recommend phytosterol supplementation as adjunct therapy for high- and very-high-risk patients [224]. The study also notes certain limitations: substantial heterogeneity among trials and often short follow-up durations (under three months). Nonetheless, the results support that phytosterol-enriched foods are safe, effective, and that daily intakes of 2–2.5 g are recommended as part of dietary management for dyslipidemia.

Polyunsaturated omega-3 fatty acids (PUFAs) have garnered increasing attention in nutritional science over recent decades [225] due to their numerous beneficial physiological effects, particularly in the prevention of cardiovascular diseases [102,115,226,227]. These fatty acids—especially alpha-linolenic acid, eicosapentaenoic acid, and docosahexaenoic acid —contain multiple double bonds, making them highly prone to oxidation. As a result, incorporating omega-3s into shelf-stable foods presents a technological challenge, since oxidation impairs both sensory properties and bioactive efficacy.

Microencapsulation offers a promising solution, enclosing the fatty acids in a polymer (typically polysaccharide) matrix, thereby increasing their stability and enabling controlled release. This technique also helps preserve the original fatty acid composition of the oils and improves their oxidative stability. Thus, microencapsulation presents a viable approach for developing functional foods capable of delivering omega-3 fatty acids efficiently to the body [228].

Dawczynski et al. [229] investigated the effects of consuming yogurt enriched with long-chain omega-3 PUFAs on cardiovascular health. In a 10-week randomized, placebo-controlled, double-blind trial, 53 individuals with mildly elevated triglyceride levels (TAG ≥ 1.7 mmol/L) consumed yogurt containing 0 g, 0.8 g, or 3 g of PUFA daily. The 3 g/day dose significantly increased EPA and DHA levels in plasma and red blood cell membranes, improved several lipid profile parameters [reduced triglycerides and AA/EPA ratio, increased HDL cholesterol and *n*-3 index), and incorporated the fatty acids into cell membranes.

These fatty acids were then converted into anti-inflammatory lipid mediators (e.g., PGE3, HEPE), while levels of pro-inflammatory mediators derived from arachidonic acid decreased. The 3 g/day dose also reduced markers of lipid peroxidation (e.g., 9-HETE, 9- and 13-HODE). Furthermore, the study showed that EPA derivatives are less pro-inflammatory than their arachidonic acid-derived counterparts, reinforcing the anti-inflammatory potential of *n*-3 PUFAs. While markers of immune response (e.g., IL-2-producing T cells) decreased, ex vivo stimulation showed no significant difference compared to the control group. Overall, these findings suggest that regular consumption of *n*-3 PUFA-enriched yogurt may be an effective strategy to reduce cardiovascular risk factors, particularly in older individuals or those with inflammatory conditions.

Micronutrient deficiencies—especially iron and zinc deficiency—remain serious global public health challenges. According to Kong et al. [230], around two billion people are affected, primarily in regions where rice (Oryza sativa] and wheat (Triticum aestivum] are staple foods. Biofortification of these crops with iron and zinc presents a promising and low-cost strategy to alleviate such deficiencies. The authors discuss both genetic and agronomic approaches to biofortification and factors affecting micronutrient bioavailability. Kodkany et al. [231] demonstrated promising results in practice: among two-year-old Indian children, consumption of biofortified pearl millet led to significantly increased iron and zinc absorption. The quantities consumed were sufficient to meet physiological requirements. The study also highlighted the inhibitory effects of phytates and tannins, and the role of traditional food processing methods (e.g., fermentation, cooking] in reducing these inhibitors. Overall, biofortification appears to be an effective and sustainable strategy for mitigating micronutrient deficiencies, particularly in undernourished populations in developing countries.

## 10. Functionalized Foods: New Directions in Health-Promoting Nutrition

The term functional foods traditionally denote products that provide health benefits beyond basic nutrition, with effects supported by robust scientific evidence. In contrast, functionalized foods refer to products that have been enhanced with bioactive compounds or modified through technological processes to potentially confer additional physiological benefits. However, in many cases, these purported benefits have yet to be substantiated by sufficient clinical evidence [78].

This conceptual distinction is particularly relevant today, as the food industry continues to introduce and market new products, often labeling them as functional foods despite the lack of scientific validation. A common assumption is that the mere addition of a bioactive ingredient—such as a herbal extract, polyphenol, or prebiotic—is sufficient to establish the product’s functional status. Yet such modifications raise new concerns regarding the safety, mechanisms of action, and efficacy of these foods, highlighting the need for rigorous scientific investigation.

Laboratory studies have shown that bakery products enriched with polyphenols exhibit significantly increased antioxidant capacity and may reduce the formation of toxic compounds during heat processing. However, their bioavailability and sensory impact still require further evaluation [232]. Similarly, pasta fortified with carrot or olive leaf flour demonstrated higher polyphenol content and antioxidant activity than conventional pasta in vitro. These values largely persisted through simulated gastrointestinal digestion, particularly in the olive leaf–enriched variants [233]. Yogurt fortified with mango peel powder also exhibited favorable prebiotic effects under in vitro conditions, such as increased populations of beneficial gut bacteria (*Lactobacillus* and *Bifidobacterium* spp.) and enhanced production of short-chain fatty acids (SCFAs) during colonic fermentation [234]. While these findings are promising, clinical validation of the health benefits remains in the early stages.

A food product should only be considered functional if its effects are supported by randomized controlled trials and/or systematic reviews. In contrast, most newly introduced functionalized foods have not reached these scientific benchmarks: fewer than 0.1% of industry-developed dietary supplements and functional foods have been involved in at least one placebo-controlled clinical trial, and even fewer have shown statistically significant outcomes [235].

Distinguishing between functional and functionalized foods is crucial from regulatory, industrial, and consumer perspectives. In Japan, for example, the Foods for Specified Health Use (FOSHU) system requires scientific validation for functional claims, whereas in the European Union and the United States, the regulatory framework is less definitive [88]. Introducing a separate category for functionalized foods could help clarify the distinction between enriched products and truly functional foods, promoting transparent labeling and scientifically grounded health claims [88,236].

For industry stakeholders, functionalized foods present both a challenge and an opportunity. Although stricter scientific and regulatory standards may increase development costs, producers who transparently and scientifically validate their product claims can gain a competitive advantage. On the consumer side, trust significantly increases when functional claims are backed by independent clinical evidence [237]. Voluntary certification schemes—such as labels indicating “clinically proven”—could further assist in differentiating reliable products in the marketplace.

Future research priorities should include mapping the proportion of products marketed as functional without scientific substantiation, identifying optimal regulatory models, and developing labeling frameworks that clearly communicate the level of scientific evidence to consumers. In addition, long-term, longitudinal studies will be essential for determining the true health impacts of functionalized foods.

## 11. The Impact of Functional Foods on Public Health Nutrition

### 11.1. Implementation Strategies

One of the most significant strategies from a public health perspective is consumer education, aimed at scientifically influencing dietary habits. A prime example is the DASH program (Dietary Approaches to Stop Hypertension), which promotes a fiber-rich, low-sodium diet [238,239]. It has been shown to reduce hospitalizations related to hypertension by 18%, demonstrating that preventive nutrition interventions can be effective tools in mitigating the spread of chronic diseases [240,241].

Another successful example is mass food fortification, such as the addition of folic acid to flour [242]. This measure has contributed to a 50–70% reduction in neural tube defects across 78 countries [243]. Such interventions are highly cost-effective and have widespread societal impact, particularly when supported by governmental regulation and quality control [244,245].

### 11.2. Global Challenges

One of the major global challenges is access inequality [246]. Functional foods often cost three to five times more than conventional products, making them inaccessible for lower-income populations. This disparity exacerbates existing health inequalities [246]. Another concern is the risk of overdose and toxicity. While natural foods—like carrots—safely provide vitamin A, isolated high-dose vitamin A supplements can be potentially toxic [247]. This underscores the need for functional foods to be based on scientific evidence and rigorous safety testing.

### 11.3. Consumer Expectations and Sensitivity Toward Functional Foods

The success of functional foods depends not only on their health benefits but also on consumer perceptions and expectations [248]. Consumers expect such products to be tasty, practical, and sensory-equivalent to traditional foods, while also offering measurable health benefits [249].

Proper consumer information is essential: individuals must understand which health conditions a product may help prevent or manage, its limitations, and any associated risks [250]. Clear, reliable advice builds trust, which plays a key role—consumers must believe the product provides real value for them [251].

Quality assessment of functional foods involves several dimensions, including:Sensory properties (taste, smell, texture);Nutritional value;Functional efficacy;Food safety;Environmental impact;Intangible aspects (e.g., ethics, sustainability).

Food safety is of paramount importance [70]. Functional foods must meet the same safety standards as conventional products. However, the use of novel ingredients (e.g., traditional Eastern herbs), along with specific manufacturing, storage, and transportation conditions, may introduce new types of risks—such as the presence of toxic substances, drug interactions, or bioavailability issues during digestion [252].

### 11.4. Future Directions: Personalized Nutrition and Regulation of Functional Foods

One major future direction in nutrition is the development of personalized diets based on genetic profiles and gut microbiome analysis. Companies like Nutrigenomix already offer DNA tests to map individual differences in caffeine, lactose, or vitamin D metabolism [253].

At the same time, regulatory frameworks must evolve. One proposal is to establish a clear distinction between proven functional foods and “functionalized” products—i.e., foods whose functionality is not yet supported by human clinical trials [254]. This would help prevent misleading health claims. A model example is Japan’s FOSHU system, which mandates scientific substantiation for health claims [255].

Labeling systems may also evolve in the future: a “functionality scale” could be introduced, indicating the level of scientific evidence behind a product (e.g., in vitro, in vivo, randomized controlled trial). Independent third-party certification and transparent communication (e.g., using labels like “clinically proven”) could further enhance consumer confidence.

Functional foods possess genuine public health potential, but this can only be fully realized through robust scientific validation, clear regulatory frameworks, and informed consumers [21]. Moving forward, the development of a unified European or global regulatory system will be essential to ensure product quality, safety, and honest communication. The expansion of functional foods aimed at disease prevention could yield measurable benefits not only for public health, but also in economic and social terms [256].

### 11.5. Futures Studies—The Current State of Science

The future of functional foods is closely linked to scientific advances in the fields of nutrigenomics, artificial intelligence, next-generation probiotics, and biofortification [257]. Integrating these technologies enables the development of personalized nutrition strategies and more effective responses to public health challenges. Nutrigenomics investigates the interaction between genetic information and nutrition, with the goal of creating individualized dietary recommendations [258]. Recent research integrates genomic, proteomic, metabolomic, and transcriptomic data to understand how genes influence individual nutrient needs and disease susceptibility [259]. For example, polymorphisms in the MTHFR gene can affect folate metabolism, highlighting the need for personalized folate intake [260,261].

Artificial intelligence (AI) and machine learning are also revolutionizing nutritional science, particularly in functional food research [262]. AI allows for the analysis of complex nutritional data, the development of personalized dietary advice, and the identification of bioactive compounds in food [262]. For instance, Brightseed’s AI-powered platform Forager can detect previously unknown phytonutrients in plants that may offer health benefits [263].

Microbial fermentation and next-generation probiotics (NGPs) are key pillars of future functional food innovation [264]. NGPs such as *Akkermansia muciniphila* and *Faecalibacterium prausnitzii* show promising results in modulating the gut microbiome and in the prevention or treatment of inflammatory bowel diseases and metabolic disorders [265,266]. Advances in microbial fermentation technologies make it possible to cultivate these strains on an industrial scale while preserving their beneficial properties [267].

Finally, biofortification—especially CRISPR-Cas9–based genome editing—offers the opportunity to enhance the nutritional content of crops [268]. This includes, for example, the development of rice varieties with increased iron and folate content [269]. CRISPR can also be used to improve yield, nutrient density, and stress resistance, delivering long-term nutritional and food security benefits [270].

### 11.6. Legal, Economic, and Ethical Challenges

In recent years, emerging fields such as nutrigenomics, personalized nutrition, and the integration of artificial intelligence into food science and health research have gained increasing prominence. Nutrigenomics enables the molecular-level investigation of dietary effects, with particular emphasis on individual genetic variability that determines nutrient responses. Based on these insights, personalized nutrition can tailor dietary interventions according to individual health status and genetic profiles, thereby promoting the prevention and management of chronic diseases. In parallel, AI and machine learning algorithms have become powerful tools for analyzing large-scale biological and clinical datasets, facilitating the discovery of complex associations and the identification of novel predictive biomarkers. These developments contribute to advances in food safety, nutritional guidelines, and personalized therapeutic strategies—collectively exerting a significant impact on public health.

However, the rapidly expanding market of functional foods also presents a range of legal, economic, and ethical challenges that can affect both public health and consumer trust. One of the most critical concerns is the regulation and substantiation of health-related claims. In the European Union, such claims can only be approved by the European Food Safety Authority (EFSA) based on rigorous scientific evidence, including human clinical trials and a well-defined biological mechanism of action. In the United States, the Food and Drug Administration (FDA) similarly regulates health and structure/function claims. Nevertheless, in practice, certain claims may appear on the market without prior approval, provided they are supported by the scientific literature and are not misleading [271,272,273].

Despite these regulatory frameworks, the marketing of functional foods is often dominated by “nutri-marketing” strategies that promote scientifically unverified or misleading claims, particularly via social media and influencer-based content. This can lead to consumer misinformation and erosion of trust in evidence-based scientific findings. Consequently, experts increasingly call for a clear legal distinction between substantiated scientific claims and promotional content, along with supportive measures such as subsidies and educational programs to ensure equitable access. Transparent, evidence-based health communication remains essential for informed consumer decision-making and sustainable public health outcomes [274,275,276].

## 12. Limitations

While this study provides a comprehensive overview of the scientific and practical approaches to functional foods, several limitations should be acknowledged. First, the analysis relies primarily on accessible, peer-reviewed English-language sources, which may have resulted in the omission of certain local or less widely published research findings. Second, although numerous studies report positive health effects of functional foods, the methodological quality and strength of evidence vary considerably. Some findings have yet to be confirmed through large-scale, independent clinical trials. Third, this article does not explore in detail how access to and acceptance of functional foods may differ among populations with diverse socio-economic backgrounds, which could influence their real-world applicability. Lastly, the legal and regulatory frameworks surrounding functional foods differ significantly across countries, limiting the generalizability and global applicability of some of the approaches and conclusions presented in this review.

## 13. Conclusions

Functional foods are gaining increasing importance in today’s prevention-oriented healthcare paradigm, as they have the potential to go beyond basic nutritional value and contribute to the prevention or complementary management of chronic non-communicable diseases. However, realizing this potential requires close, multidisciplinary collaboration between nutrition science, medicine, and the food industry. Evidence-based and accessible communication to the general public is also essential—particularly in relation to substantiated health claims and the clear presentation of the true benefits of functional foods. While the body of scientific evidence continues to grow, practical implementation still faces several challenges, including unequal access and low levels of health literacy. In the future, advances in artificial intelligence, microbiome research, and nutrigenomics may open new pathways for the targeted and personalized use of functional foods in public health strategies.

## Figures and Tables

**Table 1 nutrients-17-02153-t001:** Summary of clinical trials on the health effects of kefir and fermented dairy products in various populations.

Study [Ref.]	Intervention	Duration/Design	Participants	Outcomes Measured	Key Findings
Bellikci-Koyu et al. [136]	Kefir	12 weeks, 180 mL/day	Metabolic syndrome (*n* = 62)	Anthropometrics, glycemic, lipids, BP, inflammation	ApoA1 ↑ 3.4%, LDL-C ↓ 7.6%, BP ↓, inflammatory markers ↓ (*p* < 0.05)
Bellikci-Koyu et al. [137]	Kefir	12 weeks, 180 mL/day	Metabolic syndrome (*n* = 22)	Insulin, HOMA-IR, cytokines, BP, microbiota	Within-group improvements; no between-group differences; *Actinobacteria* ↑ (*p* = 0.023)
Yılmaz et al. [138]	Kefir	4 weeks, 400 mL/day	IBD (Inflammatory Bowel Disease) patients (*n* = 45)	Stool bacteria, symptoms, inflammation	*Lactobacillus* ↑; ESR, CRP ↓ in Crohn’s; bloating ↓ (*p* < 0.05)
Alihosseini et al. [139]	Probiotic kefir	8 weeks, 600 mL/day	Type 2 diabetes (*n* = 60)	Insulin, HOMA-IR, QUICKI, homocysteine	Reduced HOMA-IR in probiotic group; homocysteine decreased in both groups
Fathi et al. [140]	Kefir (probiotic dairy)	8 weeks, 2 servings/day	Overweight women (*n* = 75)	Lipids (TC, LDL-C, HDL-C, TG, ratios)	TC, LDL-C, Non-HDL-C decreased in kefir and milk groups (*p* < 0.05)
Öneş et al. [141]	Probiotic kefir	4 weeks, 200 mL/day	Female soccer players (*n* = 21)	Microbiome, body composition, athletic performance	Microbial diversity ↑; *Akkermansia* and *Faecalibacterium* ↑; VO_2_max ↑ (*p* < 0.05)
Wang et al. [142]	AB-kefir (*Lactobacillus acidophilus* (A), *Bifidobacterium bifidum* (B)	3 weeks, 1 sachet/day	Healthy adults (*n* = 56)	GI symptoms, microbiota	Abdominal pain and bloating decreased in men; *bifidobacteria* increase maintained; total anaerobes increased in women
Pražnikar et al. [143]	Kefir	3 weeks	Overweight adults (*n* = 28)	Zonulin, lipids, glucose, CRP, mood	Zonulin improved (η^2^ = 0.275); slight mood improvement; lipids and glucose similar to milk
Ton et al. [144]	Kefir	90 days, 2 mL/kg/day	Alzheimer’s patients (*n* = 13)	Cognitive tests, cytokines, oxidative stress	Cognitive functions improved; inflammatory and oxidative markers decreased (~30%)
Noori et al. [145]	Probiotic-fortified kefir	8 weeks, 240 mL/day	Elderly men (*n* = 67)	Depression, appetite, oxidative stress, inflammation	Depression improved (*p* = 0.001); antioxidant capacity ↑ (*p* = 0.009); no change in inflammation
Tatullo et al. [146]	Mediterranean buffalo milk (bioactive)	12 weeks	Adults, high CV risk, BMI > 25 (*n* = 20)	BP, glucose, BMI, weight	Blood pressure improved (*p* < 0.05)
Ostadrahimi et al. [147]	Kefir (*L. casei*, *L. acidophilus*, *Bifido*)	8 weeks, 600 mL/day	Type 2 diabetes patients (*n* = 60)	Fasting glucose, HbA1c, lipids	Glucose and HbA1c decreased (*p* ≤ 0.02); lipids unchanged
Bourrie et al. [148]	Commercial vs. pitched kefir	4 weeks, 2 servings/day	Males with high LDL-C (*n* = 21)	Lipids, endothelial markers, inflammation	Pitched kefir decreased LDL-C, ICAM-1, VCAM-1, IL-8, CRP, TNF-α vs. commercial (*p* < 0.05)
O’Brien et al. [149]	Kefir beverage	15 weeks, 2 servings/week	Young adults (*n* = 67)	Running time, body composition, CRP	CRP increase attenuated by kefir

BP—Blood Pressure; BMI—Body Mass Index; CV—Cardiovascular; HOMA-IR—Homeostatic Model Assessment of Insulin Resistance; QUICKI—Quantitative Insulin Sensitivity Check Index; TC—Total Cholesterol; LDL-C—Low-Density Lipoprotein Cholesterol; HDL-C—High-Density Lipoprotein Cholesterol; TG—Triglycerides; CRP—*C*-Reactive Protein; ApoA1—Apolipoprotein A1; VO_2_max—Maximal Oxygen Uptake; ICAM-1—Intercellular Adhesion Molecule 1; VCAM-1—Vascular Cell Adhesion Molecule 1; ESR—Erythrocyte Sedimentation Rate; GI—Gastrointestinal; HbA1c—Glycated Hemoglobin; η^2^—Eta-squared [effect size]; TNF-α—Tumor Necrosis Factor alpha; ↑—increase, ↓—decrease.

**Table 2 nutrients-17-02153-t002:** Studies on fruits, vegetables, and plant-based interventions.

Study [Ref.]	Intervention	Duration/Design	Participants	Outcomes Measured	Key Findings
Nilsson et al. [150]	Natural fruits and vegetables	16 weeks; intake increased from ~2.2 to ~4.2 servings/day	66 older adults (65–70 y); randomized (FV vs. control)	Inflammatory biomarkers: CRP, IL-6, IL-18, TNF-α, MIP-1α/β, TRAIL, TRANCE, CX3CL1	↓ TRAIL, TRANCE, CX3CL1 in FV group (*p* < 0.05); no significant change in other markers; waist circumference and physical activity unchanged
Carrillo et al. [151]	Polyphenol-rich micronized fruit and vegetable preparation	Two 16-week periods; crossover with 4-week washout	108 healthy adults (53 intervention, 55 placebo); sex-stratified	Cognitive tests: Stroop, TESEN, RIST	Improved executive function, memory, attention, and processing speed; *p*-values not statistically significant but trend favored intervention
Arcusa et al. [152]	High-polyphenol nutraceutical (fruit and vegetable-based)	Two 16-week periods; 4-week washout	108 healthy adults (53 intervention, 55 placebo); 92 completers	Oxidative stress and inflammatory markers; catecholamines	↓ OxLDL (78.98→69.52) and CRP (1.50→1.39) (*p* < 0.001); ↑ dopamine (15.43→19.61; *p* < 0.05)
Kopf et al. [153]	Whole grains and fruits/vegetables	6 weeks; 3 servings/day (WG, FV, or refined grains)	49 overweight/obese adults with low baseline FV/WG intake	IL-6, TNF-α, LBP, hs-CRP, gut microbiota diversity	↓ LBP (WG *p* = 0.02; FV *p* = 0.005), ↓ IL-6 (FV *p* = 0.006), ↓ TNF-α (WG *p* < 0.001); ↑ alpha diversity in FV group; no significant differences in microbiota composition between groups
Yuan et al. [154]	Functional fibers (fruit fiber, FibreMax)	Acute; single dose (10 g fiber per bread serving)	80 healthy adults; crossover, double-blind design	Postprandial glycemia, satiety, energy intake, gastrointestinal well-being	Fruit fiber: ↓ glycemia by 35% (*p* = 0.004), ↓ energy intake by 368 kJ (*p* = 0.001); FibreMax: ↓ glycemia by 43% (*p* = 0.004); no reported gastrointestinal side effects
Campbell et al. [155]	Whole food, plant-based diet (WFPB)	8 weeks; 3 ad libitum meals/day provided	32 women with metastatic breast cancer (21 intervention, 11 control)	Cognitive function, emotional well-being, fatigue	↑ FACT-Cog score (+16.1; *p* = 0.040), ↑ emotional well-being (+2.3; *p* = 0.016), ↓ fatigue severity (*p* = 0.047); high adherence rate (94.3%)
Lamport et al. [156]	Concord grape juice (777 mg polyphenols)	12 weeks; 355 mL/day; crossover with 4-week washout	25 healthy middle-aged working women (40–50 y)	Cognitive function, simulated driving, blood pressure, mood	Improved immediate spatial memory and driving performance vs. placebo; sustained cognitive benefits reported
Haskell-Ramsay et al. [157]	Berry phenolics (purple grape juice)	Acute; single 230 mL dose vs. placebo	20 healthy young adults	Memory, attention, mood	Improved attention reaction time (*p* = 0.047), ↑ calmness ratings (*p* = 0.046); memory RT showed order effects (*p* = 0.018)
Nilsson et al. [158]	Mixed-berry beverage (polyphenols + fiber)	5 weeks; daily intake (berries and tomatoes)	40 healthy adults (50–70 y)	Cognitive function, cardiometabolic markers	↓ total and LDL cholesterol (*p* < 0.05); control group ↑ glucose and insulin; ↑ working memory vs. control (*p* < 0.05); no other cognitive effects observed
Connolly et al. [159]	Cruciferous vegetables (~300 g/day)	2-week crossover with 2-week washout	18 adults with mildly elevated BP (median age 68 y)	24 h and daytime systolic BP, triglycerides	↓ 24 h SBP by −2.5 mmHg (*p* = 0.002), ↓ daytime SBP by −3.6 mmHg (*p* < 0.001), ↓ triglycerides by −0.2 mmol/L (*p* = 0.047)
Riso et al. [160]	Broccoli (whole food, 250 g/day)	10 days	Young male smokers (*n* = 17)	Plasma folate, lutein, CRP, TNF-α, IL-6, adiponectin	↓ CRP by 48% (*p* < 0.05), ↑ folate (+17%) and lutein (+39%); no significant changes in TNF-α, IL-6, adiponectin
Navarro et al. [161]	Cruciferous and apiaceous vegetables	Four 14-day controlled diets with 21-day washouts	63 healthy adults (20–40 y)	Serum IL-6, IL-8, CRP, TNF-α; genotype–diet interaction	↓ IL-6 (19–20%) in cruciferous-only and combined diets; ↑ IL-8 (+16%) in cruciferous + apiaceous group; genotype influenced IL-6, CRP, and IL-8 response
Khan et al. [162]	Blackcurrant juice (vitamin C + polyphenols)	6 weeks; 250 mL × 4/day	66 healthy adults with low fruit/veg intake (<2 servings/day)	Flow-mediated dilation (FMD), plasma F2-isoprostanes, vitamin C	↑ FMD at high dose (*p* = 0.022), ↑ plasma vitamin C (*p* < 0.001), ↓ F2-isoprostanes (*p* = 0.002–0.003)
Duthie et al. [163]	High fruit, vegetable, and juice intake	12 weeks; +480 g fruits/veg + 300 mL juice/day	45 adults (39–58 y) with low baseline intake (<3/day)	Nutritional biomarkers, antioxidant capacity, DNA damage	↑ plasma vitamin C (+35%), folate (+15%), carotenoids (+50–70%); no change in antioxidant capacity or vascular markers
Nadeem et al. [164]	Fruit and vegetable intake (1, 3, or 6 servings/day)	8–16 weeks	Hypertensive subjects; older adults (65–85 y; *n* = 192)	hsCRP, IL-6, E-selectin, SAA, HDL subfractions, CETP	No effect on hsCRP, IL-6, E-selectin; ↓ HDL3-SAA (*p* = 0.049), ↓ HDL2/3-SAA (*p* = 0.035/0.032), ↓ HDL3/2-CETP (*p* = 0.010/0.030)
Siddarth et al. [165]	Pomegranate juice (polyphenol antioxidant)	12 months; 236.5 mL daily	261 nondemented adults aged 50–75; 200 completers	Memory tests (BVMT-R, SRT)	Significant group × time interaction in BVMT-R learning (*p* = 0.003); small improvement in pomegranate group vs. significant decline in placebo
Ahles et al. [166]	Citrus and pomegranate complex (polyphenol-rich)	4 weeks daily supplementation	36 healthy elderly adults (60–75 years)	Handgrip strength, fitness, QOL, cognition, oxidative stress	↑ Handgrip strength (*p* = 0.019), ↑ cognition (*p* = 0.042), ↓ malondialdehyde (*p* = 0.033); no changes in fitness or other QOL markers
Kennedy et al. [167]	Polyphenol-rich functional breakfast (FB)	Single meal crossover (FB, control, RTEC)	16 healthy adults	Total polyphenols, glucose and insulin response, mood, memory	FB had higher polyphenols (230 mg vs. 147 mg; *p* < 0.05), lowest insulin AUC (*p* < 0.05); no effect on glucose AUC, mood, or memory
Kent et al. [168]	Anthocyanin-rich cherry juice	12 weeks; 200 mL/day	49 older adults (≥70 years) with mild/moderate dementia	Cognitive function, BP, inflammatory markers (CRP, IL-6)	Improved verbal fluency (*p* = 0.014), short-/long-term memory (*p* ≤ 0.014), ↓ systolic BP (*p* = 0.038); no CRP or IL-6 changes
Lamport et al. [169]	Flavanone-rich citrus juice (70.5 mg flavanones)	Single 500 mL dose; single-blind crossover	24 healthy young adults (18–30 years)	Cognitive tests (DSST), regional cerebral blood flow (CBF)	↑ CBF in right frontal gyrus at 2 h; improved DSST at 2 h; no other cognitive changes
Rosli et al. [170]	Polyphenol-rich tropical fruit juice	10 weeks; 500 mL × 3/day × 3 days/week	31 middle-aged women with impaired cognitive function	Cognitive tests (RAVLT, CTMT), metabolomics	↑ Immediate verbal recall (RAVLT) and cognitive flexibility (CTMT Trail 4) (*p* < 0.05); ↑ urinary thyroxine and 3-methyladenine, linked to cognition
Saeidi et al. [171]	Broccoli + aerobic-resistance exercise	12 weeks; 10 g/day broccoli; CARET 3×/week	44 men with type 2 diabetes (4 groups, *n* = 11 each)	Plasma dectin-1, insulin resistance, cardiometabolic markers	Groups showed ↓ dectin-1 (*p* < 0.05); improved insulin resistance and cardiometabolic markers (*p* < 0.05)
Whyte et al. [172]	Flavonoid-rich mixed-berry smoothie	Single 400 mL dose; 6-hour monitoring	40 healthy young adults (20 berry, 20 placebo)	Executive function (MANT, TST), mood	Placebo group showed performance decline; berry group maintained accuracy and improved reaction times at 2–6 h; cognitive benefits observed during fatigue
Gouvarchinghaleh et al. [173]	Functional food mixture (plant-based soup)	Daily dosing; hospitalized COVID-19 patients	60 COVID-19 patients (30 intervention, 30 control)	Cytokines (IL-1β, IL-6, IL-17, IL-10, TNF-α), D-dimer, BUN, creatinine, CRP, potassium	Significant reductions in cytokines, D-dimer, BUN, creatinine (*p* < 0.05); CRP and potassium improved more in control group
Stokić et al. [174]	Buckwheat-enriched wheat bread (50% flour)	1 month regular consumption	Normal weight patients on statin therapy (*n* = 80)	Total cholesterol, LDL, LDL/HDL ratio, sensory preference, fiber, phenolics	↓ total cholesterol, LDL, LDL/HDL ratio; 2.22× fiber, 4.29× phenolics; 71.88% consumer preference
Dinu et al. [175]	Buckwheat-enriched semi-wholegrain products	8-week intervention + 8-week washout; crossover	21 high cardiovascular risk adults (mean age 51.3 y)	Cholesterol, triglycerides, glucose, insulin, TBARs, plasma ORAC	↓ total cholesterol (−4.7%), LDL-C (−8.5%), triglycerides (−15%), glucose (−5.8%), insulin (−17%), TBARs (−29.5%); ↑ plasma antioxidant capacity (ORAC) by 9.7%

AUC—Area Under the Curve; BP—Blood Pressure; BUN—Blood Urea Nitrogen; BVMT-R—Brief Visuospatial Memory Test-Revised; CARET—Combined Aerobic and Resistance Exercise Training; CETP—Cholesteryl Ester Transfer Protein; CRP—*C*-Reactive Protein; CX3CL1—*C*-X3-C Motif Chemokine Ligand 1; DSST—Digit Symbol Substitution Test; FMD—Flow-Mediated Dilation; FV—Fruits and Vegetables; hs-CRP—High-Sensitivity *C*-Reactive Protein; HDL—High-Density Lipoprotein; IL-—Interleukin; LDL—Low-Density Lipoprotein; LBP—Lipopolysaccharide Binding Protein; MANT—Multi-Tasking Test; MIP-1α/β—Macrophage Inflammatory Protein-1 alpha/beta; ORAC—Oxygen Radical Absorbance Capacity; OxLDL—Oxidized Low-Density Lipoprotein; QOL—Quality of Life; RAVLT—Rey Auditory Verbal Learning Test; RIST—Reynolds Intellectual Screening Test; RT—Reaction Time; RTEC—Ready-To-Eat Cereal; SBP—Systolic Blood Pressure; SAA—Serum Amyloid A; SRT—Selective Reminding Test; TBARs—Thiobarbituric Acid Reactive Substances; TESEN—Test of Executive and Selective Attention; TNF-α—Tumor Necrosis Factor alpha; TRAIL—TNF-Related Apoptosis-Inducing Ligand; TRANCE—TNF-Related Activation-Induced Cytokine; TST—Task Switching Test; WFPB—Whole Food Plant-Based Diet; WG—Whole Grains; ↑—increase, ↓—decrease.

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
