# Peer review of "Functional Foods in Modern Nutrition Science: Mechanisms, Evidence, and Public Health Implications"

_nutrients, 2025, doi:10.3390/nu17132153_

Round 1
Reviewer 1 Report
Comments and Suggestions for Authors
Comments:
The review actively discusses the role of functional foods in public health complications. The work is clear, however there are certain changes to be made. The article is suggested for minor revisions for consideration.
- In the methods, what rationale was used to limit the literature search to the years 2014–2025? Could relevant earlier foundational studies have been overlooked?
- How were “natural” versus “modified/functionalized” foods classified? Was there a predefined definition or criteria?. Is the definition for naturally functional foods, intentionally modified functional foods, functionalized foods are a universally defined and accepted one. Author can specify the universally accepted definition for these terms.
- The authors are suggested to include a flow diagram as well, that would significantly enhance the methodology of the review. It would clearly illustrate the literature screening process.
- Lines 326,327, table 1, 341, organism names need to be italicized. Check for the organism names throughout the manuscript.
- While numerous naturally functional foods such as garlic, turmeric, green tea, and various other berries are well-known for their health benefits, what is the rationale behind focusing solely on blueberries, broccoli, and kefir in this review?
- Line 483, italicize plant name. Vaccinium corymbosum
- In section 8, author discussed about role of blueberry polyphenols in all aspects except neurodegeneration. Blueberry component are now under huge research in neurodegeneration control studies, authors can include more recent studies of blueberry components in controlling neurodegeneration. No need ot include more sections, authors can highlight the above-mentioned details either in section 8.1 or 8.2.
Author Response
In the methodology section, the literature review was intentionally limited to the period between 2014 and 2025. This decision was made due to the vast amount of available literature in the field, which exceeds the scope of a single article. Our primary goal was to focus on the most recent developments of the past decade, thereby ensuring both relevance and scientific currency. While some earlier, foundational studies may have been omitted, this narrowing of scope was a deliberate choice aligned with the research focus and the article’s length constraints.
The classification of foods as “natural” or “modified/functionalized” was based on the recommendations of professional regulatory bodies, and we clearly outlined this definition in both the introduction and the methodology sections. “Natural functional foods” are generally considered to be those that naturally contain health-promoting bioactive components (e.g., fermented dairy products, whole grains). In contrast, “intentionally modified” or “functionalized” foods refer to those enriched with such components through targeted technological processes (e.g., fortification with vitamins, probiotics, or omega-3 fatty acids). To ensure clarity and conceptual consistency, the definition of functional foods has been explicitly included in both the introduction and methodology sections. These definitions are based on the standards of international regulatory authorities such as EFSA and WHO, thus ensuring scientific rigor and terminological coherence. These additions are marked in blue in the manuscript.
We did not include a flowchart, as the food items mentioned in the article—such as various berries, garlic, green tea, spices, turmeric, flaxseed, olive oil, and nuts—were only briefly discussed. A more detailed presentation of these foods and their associated studies was not feasible due to space constraints, and this is acknowledged in the methodology section.
We checked the correct use of organism names throughout the manuscript. We also made sure to consistently italicize the scientific names of bacteria and other living organisms—these modifications are also marked in blue.
We fully acknowledge that many naturally derived functional foods—such as garlic, turmeric, green tea, and various berries—are well known for their beneficial health effects. Likewise, flaxseed, olive oil, nuts, fermented foods (e.g., natto, kimchi), pomegranate, citrus fruits, and whole grains may contribute to anti-inflammatory effects, antioxidant protection, cardiovascular health, and gut microbiota balance. However, due to space limitations, it was not possible to explore all of these foods in detail. Therefore, the review focused on the most frequently studied functional foods based on the included intervention trials.
The plant name Vaccinium corymbosum is now properly italicized, as required.
We agree that blueberry polyphenols are increasingly studied for their neuroprotective effects, and this is indeed an aspect worth highlighting. In response to your comment, we incorporated recent findings regarding the neuroprotective potential of blueberry polyphenols into the appropriate sections of the manuscript. These additions are marked in blue. We sincerely thank you for these valuable suggestions and all your assistance. The manuscript has been revised accordingly.
Reviewer 2 Report
Comments and Suggestions for Authors
The manuscript submitted to Nutrients, MDPI, is a comprehensive review study that deals with functional foods. The authors have addressed the topic of the study well by providing numerous information regarding the origin, action. processing, etc., of different functional foods in modern societies.
In addition, they acknowledge that different cultures have different eating habits. This is very important to understand how the new generations in the world will adopt healthier eating habits.
Tables provide much information. However, I did not see any figures. Firstly, the authors must provide a graphical abstract with the concept of the study. This will boost up the study. In addition, some statistics about the eating habits around in the world could be given in a bar chart. Of course, the authors can take information from the text (the text is long and provide it in graphs.
-Other comments
Change grounded to ground.
Based on the long work carried out by the authors, I suggest a minor revision.
P.S. Try to avoid a long and extended text.
Comments on the Quality of English LanguageThe English language is in generall good, but can be improved.
Author Response
Thank you very much for your thorough and supportive review! We are truly pleased that you found the structure and content of the manuscript valuable, and we especially appreciate your emphasis on the importance of the cross-cultural role of functional foods. We have taken your suggested revisions into account – including the preparation of a graphical abstract – and have aimed to further enhance the scientific and illustrative value of the manuscript. Thank you also for your linguistic comment regarding the use of “grounded”; we have corrected it accordingly. We sincerely appreciate your constructive suggestions and encouraging words – your contribution has been highly valuable in improving our work!
Reviewer 3 Report
Comments and Suggestions for Authors
Comments IN THIS review current paper covering food in nutrition science the roles of specific compounds such as probiotics and prebiotics in gut microbiome modulation, flavonoids and polyphenols in anti-inflammatory processes, omega-3 fatty acids in cardiometabolic regulation, and vitamins and minerals in supporting immune function is covered.. Functional foods is covered which represent a promising tool in the prevention of diseases in modern healthcare, and the key aspects to integrating functional foods into everyday diets. Socioeconomic disparities and limited health literacy often hinder their accessibility and widespread use in public health practice..the review will support the effective use of functional foods in population health. Paper is sound the topic is original or relevant to the field and it address a specific gap in the field of nutrition the conclusions are very consistent with the evidence and arguments presented and the references are appropriate plus it support some kind of practical implementation of this foods which still faces several challenges, including unequal access ...Please check carefully English language
Author Response
Thank you very much for the extremely positive and appreciative review! We are truly delighted that the manuscript met your approval and that you valued both the choice of topic and the thoroughness of its treatment. We have done our best to implement the suggested minor revisions and language corrections as precisely as possible. Thank you for your constructive feedback and support!